# Development and Properties of New Mullite Based Refractory Grog

**DOI:** 10.3390/ma14040779

**Published:** 2021-02-07

**Authors:** David Zemánek, Karel Lang, Lukáš Tvrdík, Dalibor Všianský, Lenka Nevřivová, Petr Štursa, Pavel Kovář, Lucie Keršnerová, Karel Dvořák

**Affiliations:** 1Faculty of Civil Engineering, Brno University of Technology, Veveří 331/95, 602 00 Brno, Czech Republic; zemanek.d@fce.vutbr.cz (D.Z.); nevrivova.l@fce.vutbr.cz (L.N.); 2P-D Refractories CZ JSC, Nádražní 218, 679 63 Velké Opatovice, Czech Republic; Karel.Lang@pd-group.com (K.L.); Lukas.Tvrdik@pd-group.com (L.T.); petr.stursa@hotmail.cz (P.Š.); 3Department of Geological Sciences, Faculty of Science, Masaryk University, Kotlářská 267/2, 602 00 Brno, Czech Republic; dalibor@sci.muni.cz (D.V.); Pavel.Kovar@pd-group.com (P.K.); Lucie.Kersnerova@pd-group.com (L.K.)

**Keywords:** mullite, refractory, high-alumina grog, kaolin, claystone

## Abstract

The presented study is focused on optimization and characterization of a high-alumina refractory aggregate based on natural raw materials—kaolins, claystone, and mullite dust by-product (used to increase the alumina and mullite contents, respectively). In total, four individual formulas with the Al_2_O_3_ contents between 45 and 50 wt.% were designed; the samples were subsequently fired, both in a laboratory oven and an industrial tunnel furnace. The effects of repeated firing were examined during industrial pilot tests. Mineral and chemical compositions and microstructures, of both the raw materials and designed aggregates, were thoroughly investigated by the means of X-ray fluorescence spectroscopy, powder X-ray diffraction, and optical and scanning electron microscopies. Porosity, mineral composition, and mullite crystal-size development during the firing process were also studied. Based on the acquired results, the formula with the perspective to be used as a new mullite grog, featuring similar properties as the available commercial products, however, with reduced production expenses, was selected. The quality of grog determines to a large extent the properties of the final product. Hence, optimization of aggregates for specific refractories is of a great importance. The production of engineered aggregates provides the opportunity to utilize industrial by-products.

## 1. Introduction

Although numerous scientific papers dealing with ceramics and refractories have been published, few authors have focused on refractory grog so far. Moreover, in refractories engineering, the research has mostly been focused on the matrix; the attention has shifted to aggregates only recently [1]. Aggregates are an indispensable part of refractories, they enhance the overall properties, such as volume stability during firing and high temperature behaviour, and represent 50–100% of the raw mix of most final products. The production of refractories is, among other factors, controlled by the availability of grog [2,3]. Engineering of aggregates (i.e., the entire production system, including design, customization, properties testing, evaluation of performance of refractory products, and their application for desired purposes) is expected to bring new breakthroughs in refractories technologies. Design and customization of aggregates comprise designing of shape, surface properties, chemical and phase composition, as well as optimizing microstructure [1]. In the past, refractory materials at end-of-life were dumped in landfills and new bricks were made of new raw materials. Such wasting had a huge impact on the environment, especially because of the continuous mining of raw materials [4,5]. This attitude has changed not only for ecological, but also for economic reasons [6,7].

The general situation in the refractory industry has changed significantly during the last decades (especially in recent years), which also affected the production of fired clay. Along with drop in demand for standard fired clay caused by changes in the steel manufacturing technology, the production of fired claystone has decreased. The enormous changes worldwide were primarily given by the establishment of new ecological regulations in China, exhaustion of claystone deposits in Europe (e.g., France, Poland, Czech Republic), and lack of other raw materials used for grog production, such as magnesite, bauxite, and all types of aluminium oxides. Limited claystone resources led to the heap utilization and briquetting technology application, but the results of these approaches were unsatisfactory regarding shaping and firing. These were the impulses beyond the development of high alumina refractory grog [8,9,10].

Modern temperature-stable components hot-produced from metallic materials based on elements such as titanium [11], rare-earth metals [12], or tungsten [13] introduce the necessity to develop durable temperature-resistant appliances for their production and processing. To resist high temperatures and maintain the shape and functionality, the interior of virtually any furnace is made of refractory materials, which have become sophisticated products with carefully designed compositions. The ceramic industry constantly develops methods to lower the cost and increase the quality and final properties of the materials [14,15]. 

Classification of refractories according to the chemical composition is based on the anion: cation ratio and consists of three types of materials: acidic, basic, and neutral. The anion: cation ratio higher than 1.5:1.0 indicates acidic refractory materials, for example SiO_2_ containing one cation and two anions. The neutral ratio is equal to 1.5:1.0, for example Al_2_O_3_, and 1.0:1.0 indicates basic refractories, for example MgO. This classification is widely used in the metallurgic production because the refractories have to be compatible with alloys processing. Another classification can be, for example, by methods of installation—shaped and unshaped, by methods of manufacture—fused and sintered, and by porosity content—porous and dense [14,16]. 

Each type of refractory material, including grog, is applied in different industrial sectors given by their characteristic properties. Aluminosilicate refractories, mullite materials, and fireclay belong among acidic up to neutral refractories. They are usually applied in electric furnaces, coke ovens, annealing furnaces, etc. They contain silica and between 40 to 90 wt.% of alumina. The dominant raw materials for these types of refractories are clays [4,14,17].

The final intermediate phase of the sintering process is mullite. The earliest synthesis of mullite was done by firing kaolinite. During heating of kaolinite, X-ray amorphous metakaolinite is produced at first. Subsequently, so called spinel phase and γ-alumina are formed, and finally, above 1000 °C, mullite and amorphous silica are produced. To avoid the formation of amorphous silica, additional alumina has to be added to the initial raw material [18]. The chemical composition of mullite in refractories is usually close to 3Al_2_O_3_·2SiO_2_, which corresponds to the Al_2_O_3_ content of nearly 72 wt.%. The idealized formula of the most commonly produced mullite is 3Al_2_O_3_·2SiO_2_, however, the ratio of alumina and silica may vary from 2:1 to 3:2. Among the specific properties of mullite are high thermal resistance up to 1700 °C in air atmosphere, low thermal expansion coefficient 6 × 10^−6^ K^−1^ and conductibility 4–6 W/(m K) at 100–1400 °C, high creep and corrosion resistance, and, last but not least, favourable physical and mechanical behaviour [18,19]. The firing process is realized in shaft, tunnel, or rotating kilns at the maximum temperature usually not exceeding 1350 °C. In principle, artificial mullite can be obtained in two ways. The majority of mullite in industrial conditions is produced via solid-state synthesis by firing clay, Al_2_SiO_5_ minerals (andalusite, sillimanite, and kyanite), bauxite, and other solid raw materials. Another way is the sol-gel synthesis. Solution-sol-gel-derived mullites are characterized as chemical-mullites. They are synthesized by chemical reaction, pyrolysis, and mullitization [18,20]. Each of these ways of mullite formation provides mullite with different properties [16].

There are two ways of industrial production of materials with high mullite content (above ~50% of mullite): sintering (>1500 °C), and fusing (>1830 °C, in some cases even >2000 °C). The term sinter-mullite refers to mullites synthesized by heating to temperatures below the melting point (to crystallize and densify the mullite). Fused-mullites are prepared by heating alumina and silica mixtures to temperatures above the melting point, followed by cooling, during which mullites crystallize. Mullites produced by fusing have higher Al_2_O_3_:SiO_2_ ratios than those produced by sintering from identical initial materials. The amount of alumina in sinter-mullites usually does not exceed 77 wt.% [3,21]. Mullite contents in fired clays or claystones are usually below 65 wt.% of the whole material, including the amorphous phase. Extreme mullite contents occur in so called fused mullites, which may be used as components of high alumina grogs to enhance their positive properties. From the chemical point of view, positive performance of mullite based grog is determined by high content of alumina and low contents of CaO, MgO, Fe_2_O_3_, and alkalis. The presence of detrimental elements, mainly alkalis, may significantly decrease the overall heat fatigue resistance [18,22]. 

The paper presents the engineering, characterization, and manufacturing possibilities for new mullite based aggregates with Al_2_O_3_ content between 45 and 50 wt.%. The main objective of the research is to present new aggregates featuring properties similar to those of commercially available aggregates, however, this solution is much cheaper since we use claystone and kaolins mixed with fused mullite by-product.

## 2. Materials and Methods

### 2.1. Raw Materials and Mixes

Raw mixes with the contents of Al_2_O_3_ between 45 and 50 wt.% were designed and prepared in industrial conditions (P-D Refractories CZ JSC, Velké Opatovice, Czech Republic). To reach such high alumina contents, supporting raw materials were necessary to be added to the natural claystone and floated kaolins (Table 1 and Table 2). For economic reasons, MOTIM White Fused Mullite produced by Electrocorundum Ltd., Mosonmagyaróvár, Hungary, was chosen (0–1 mm fraction; see Table 3 for the particle size distribution). MOTIM mullite is a by-product from mechanical processing of fused mullite blocks designed for glass industry applications. Therefore, it can be considered as a secondary raw material. The ratio of 85% claystone–kaolin and 15% MOTIM mullite was used to reach the Al_2_O_3_ content of at least 45%. For the chemical and phase compositions of the raw material see Table 4 in the Results section. 

For the pilot industrial experiments, pressed briquettes from the raw mixes with the approximate dimensions of 7 cm × 5 cm × 3 cm were prepared. The briquetting technology enables preparation of bodies of optimized sizes and positively impacts reduction in wasting of raw materials (the reserves of kaolins and claystones are limited). For laboratory experiments, the mixes were additionally homogenized with a wheel mixer for 10 min. Before pressing with the pressure of 10 MPa into cylinder bodies with 5 cm in diameter, 10% of water was added to the dry mixes.

Three firing experiments were used:Firing in laboratory oven—heating 10 °C/min to the maximum temperatures of 1250, 1450, and 1550 °C, followed by 5 h soaking at the maximum temperatureOne cycle of firing in industrial tunnel kiln: 1480 °C with 5 h soakingTwo cycles of firing in an industrial kiln with the regime identical as listed above (ad 2)

### 2.2. Analytical Methods

Chemical composition analysis of the raw materials was performed by wavelength-dispersive X-ray spectroscopy (WDXRF) using SPECTROSCAN MAKC-GV instrument (Spectron Company, St. Petersburg, Russia) equipped with QUANTITATIVE ANALYSIS 4.0 software. Samples were analysed in forms of fused beads.

Powder X-ray diffraction analysis of the raw materials and engineered aggregates was conducted on Panalytical Empyrean diffractometer (Malvern Panalytical Company, Almelo, The Netherlands) equipped with Cu-anode, 1-D position sensitive detector at convention Bragg–Brentano reflection geometry. The setting were step size–0.013 2θ, time per step—188 s, and angular range 5–80 2θ. Contents of the amorphous phase were quantified using the addition of 10 wt.% fluorite (CaF_2_) as an internal standard. Quantitative phase analysis was done via the Rietveld method using Panalytical High Score 3.0 plus software.

Polarizing light microscopy (PLM) examination was performed on 30 μm thick polished thin sections using an Olympus BX 51 microscope (Olympus Company, Tokyo, Japan). Scanning electron microscopy with X-ray microanalysis (SEM/EDS) was conducted on gold-coated mechanically broken specimens (for morphological analyses) and on polished carbon-coated thin sections (for chemical microanalyses) using TESCAN MIRA 3 (Tescan Company, Brno, Czech Republic) instrument with the accelerating voltage of 30 kV.

Determinations of bulk density, apparent density, apparent porosity, and water absorption were done by the well-known hydrostatic gravity method (e.g., ČSN EN 993-1 standard [23]). Apparent porosity and pore size distribution was determined by mercury porosimetry using Thermo Finnigan POROTEC Pascal 140–240 instrument (ThermoFisher Scientific, Waltham, MA, USA).

## 3. Results and Discussion

### 3.1. Characterization of Raw Materials

The alumina content (between 42–44 wt.%) and alumina:silica ratio (0.77–0.81) was similar for the claystone (W-super) and two of the kaolin samples (KN-83 and GP3). In DS1 kaolin, both the parameters were significantly lower (37.93 and 0.67%). The Al_2_O_3_:SiO_2_ ratio in MOTIM mullite grog (3.26) was much closer to the mullite value of 2:1 (3.39), than 3:2 (2.54), which corresponds to the production by fusing. The Fe_2_O_3_ content was between 1.03 (claystone) and 0.43 (KN-83), while in the MOTIM mullite it reached only 0.01%. The CaO contents were below 0.4% in all the clay and kaolin samples. The contents of alkalis in all the raw materials were relatively low with the exception of DS1 kaolin, where the content of K_2_O reached 3.53%.

The contents of minerals present in the raw materials reflected their chemical compositions: the highest kaolinite content (over93%) was identified in the KN-83 sample, and the lowest one (below 70%) in the DS1 kaolin. The most diverse association of clay minerals was found the W-super claystone, where, besides kaolinite and illite, smectite was also present. The highest contents of “impurities”, quartz and feldspars, were found in the DS1 kaolin. The high percentage of feldspars corresponds to the high contents of potassium in this sample. Chemical and phase compositions of the used raw materials are given in Table 4.

Morphology of the raw materials was observed using SEM. The size of kaolinite pseudohexagonal platelets normal to (0 0 1) was apparently the largest (up to 5 μm) in the DS1 kaolin, where also the crystal size was the most uniform. The smallest kaolinite crystals (<1 μm) were observed in W-super claystone, and GP3 kaolin samples. It can be assumed that kaolinite crystal size, its distribution and, hence, specific surface may affect reactivity. For comparison of microstructures see Appendix A. The SEM observation of MOTIM mullite (Figure A5) corresponds to the sieve analysis results. The particle size of the MOTIM supporting raw material was below 1 mm with the mean value of 0.28 µm (Table 3).

### 3.2. Tailored Aggregates

#### 3.2.1. Phase Composition

The bulk phase composition, including amorphous (“glassy“) phase, of all the aggregates prepared in both the laboratory and industrial conditions was analysed by XRD (Table 5, Appendix B). The cristobalite and amorphous phase contents of well homogenized samples fired in a laboratory oven differed significantly (Figure 1). In aggregate samples K and D, cristobalite did not appear at any of the firing temperatures (1250, 1450, and 1550 °C). The content of the amorphous phase was relatively high already at 1250 °C and increased negligibly with increasing temperature. For sample D, such a severe melting already at lower temperatures can be explained by the abundance of potassium in the formula (2.89%, Table 2). However, for sample K, the chemical composition did not provide any clear explanation of the course of the melting process. In samples W and G, the contents of the amorphous phase and cristobalite correlated negatively with each other. Cristobalite melted below 1450 °C in sample W, and above this temperature in sample G. Residual quartz in the amounts above 1% remained in samples W and G at 1450 °C. On the other hand, quartz did not transform to cristobalite but melted directly in samples K and D.

Mullite contents did not significantly change between 1250 and 1550 °C in the laboratory-prepared samples (Figure 2). In the samples fired at 1250 °C, both the structures of mullite—close to 3:2 and 2:1—were present. At higher temperatures, the 3:2 structure became dominant. Based on the mullite peak profiles, it is apparent that the crystallinity of mullite increased between 1250 and 1450 °C. The mullite peaks’ values of full width at half maximum (FWHM) were significantly higher in the samples fired at 1250 °C compared to those exposed to higher temperatures.

After firing at an industrial tunnel kiln, mullite content was similar as in analogous laboratory-prepared sample for sample G. In samples W and D, the acquired mullite contents were lower, and in sample K it was higher (Figure 2). Mullite contents in the industrial samples can be affected by inhomogeneity of the raw mix. The higher mullite content in sample K correlated with the low amorphous phase content (below 10%), in comparison to the same formula fired in laboratory conditions (over 36–44%). It may be assumed that secondary mullite formed by crystallization of the melt during slow cooling in the tunnel kiln. In all the formulas, mullite content slightly decreased after the second cycle of industrial firing, which may be attributed to corrosion by melt. Fireclay refractories are usually not exposed to such high temperatures (1550 °C) during their lifecycle. The common maximum exposition temperature does not exceed 1400 °C. Therefore, the risk of the occurrence of corrosion by melt during their working lifetime is limited.

Table 5 shows the effect of the firing temperature on the phase composition of laboratory-prepared samples, the comparison with industrial samples produced by one and two firing cycles, and the reference material. The phase composition of sample K fired in the industrial tunnel kiln was the closest to the commercial refractory mullite aggregate used as the reference material. In laboratory conditions, similar composition was acquired for the sample with formula G (Table 5).

The heterogeneous microstructure of the industrially prepared samples was observed by polarizing light microscopy (PLM). Physical-mechanical properties of the composite products are usually strongly affected by the interfaces of different used materials. Fired clay nodules, locally of the size over 1 cm, were surrounded by mixtures of well-dispersed clay binder and MOTIM mullite particles in all the examined samples (for examples see Figure 3). The interfaces of these regions were relatively sharp (Figure 4) with no substantial porosity possibly decreasing the physical-mechanical properties of the samples. Grey colouring of the nodules near the interfaces was caused by migration of iron oxides towards their central parts. In some cases, iron oxides formed isolated circular spots in the fired clay nodules. The rims of MOTIM mullite did not exhibit any severe corrosion (Figure 3). 

#### 3.2.2. Mullite Crystal Size Development

The development of mullite crystals‘ mean size along c crystallographic axis with firing temperature (5 h soaking) in the clay part of the grog was observed and calculated using SEM (Figure 5; mullite was identified by EDS analysis). Similar trend was observed in all the samples (Table 6). The slowest growing of mullite crystals was observed in DS1 kaolin, which was the most alkali-rich one. However, the values were similar to each other (in comparison to commercial samples, for which they are approximately by one order of magnitude larger). 

#### 3.2.3. Pore Size Analysis

The effect of firing temperature on pore-size distribution development in laboratory-prepared samples is shown in Figure 6 and Figure 7. The total volume of open pores with the diameters below 1 µm was significantly lower in all the samples fired at 1450 °C (in comparison to those fired at 1250 °C). The total volume of small pores also decreased between 1450 and 1550 °C in the G-L and W-L samples. In the K-L and D-L samples, slight increase in open porosity between 1450 and 1550 °C can be observed. The laboratory-prepared grog samples based on kaolins featured lower total volumes of pores with the diameters between 1 and 40 µm than the sample based on claystone (W-L), this pore size for which was dominant. After firing at 1250 °C, all the kaolin-based samples displayed similar pore-size distributions with approximately 85 vol.% of pores below 1 µm in diameter. The lowest total porosity after all the firing regimes was observed in D-L sample. The increase in the firing temperature from 1250 to 1450 °C led to significant clinkering of K-L and D-L samples, which manifested by the dominance of small pores (below 1 µm).

#### 3.2.4. Comparison of Production Expenses of Tailored Aggregate with Commercial Products

The production expenses for the refractory grog produced by the pilot industrial experiments were compared to the aggregates available on the market (Table 7). The tailored aggregate prices were calculated according to the equation in Scheme I. They include the price of raw material and all the procedures—processing, mixing, shaping, firing, crushing, and sorting. Therefore, the final cost can be considered the basis for the future commercial price. As can be seen in Table 7, the production expenses were considerably lower than those of the commercially available aggregates.

Scheme I: Equation of the price calculation.
(1)$$P=TL\times \left[\left(I_{x}+M\right)\times LOI)+Mi+F+C+S_{o}+S_{e}\right]\times \mathrm{Cpi}$$
*P* = total price in €, *TL* = technological loss, *LOI* = loss of ignition, CP = contribution profit price of: *I_x_* = raw materials (kaolin or claystone), *M* = mullite dust, *Mi* = mixing, *F* = firing, *C* = crushing, *S_o_* = sorting, *S_e_* = sieving

## 4. Conclusions


Tailoring the aggregates is necessary to achieve positive properties of modern refractory materials.Available clays of central European origin and MOTIM mullite by-product were tested as raw materials for mullite grog production.In total, two sets of experiments were conducted: laboratory experiments and pilot industrial tests. The observed differences of phase and microstructure properties of analogical formulas were caused by different firing conditions, as well as inevitable inhomogeneity of samples produced in industrial conditions.The applicability of clays for refractory aggregates production is limited by contents of alkalis.Repeated firing in a tunnel kiln with the maximum temperature of 1550 °C had negative impact on phase composition. The content of mullite slightly decreased in all the four investigated formulas. However, the firing temperature was much higher than the assumed exposition temperatures.Phase compositions and microstructures, including open porosity, of the products were compared to those of commercially available mullite-based aggregates. The properties of the aggregate with KN-83 kaolin matrix fired in an industrial tunnel kiln were comparable to the reference material.In comparison to similar aggregates available in the market, remarkably cheaper production costs were estimated. Further research on this aspect is needed to clarify the extent of the production savings.


## Figures and Tables

**Figure 1 materials-14-00779-f001:**
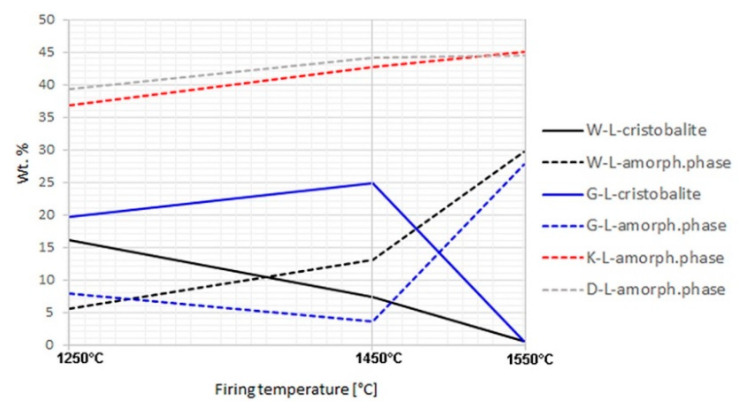
Cristobalite and amorphous phase contents in laboratory-fired samples; no cristobalite was detected in K-L and D-L samples.

**Figure 2 materials-14-00779-f002:**
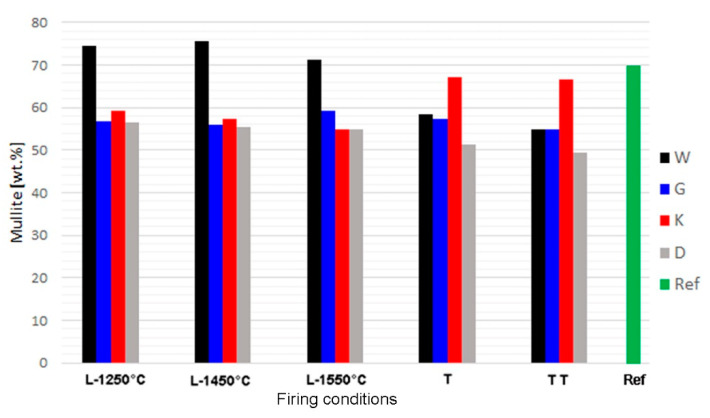
Contents of mullite in synthetised samples and reference material; L = l aboratory firing, T,TT = one and two cycles of firing in industrial tunnel kiln.

**Figure 3 materials-14-00779-f003:**
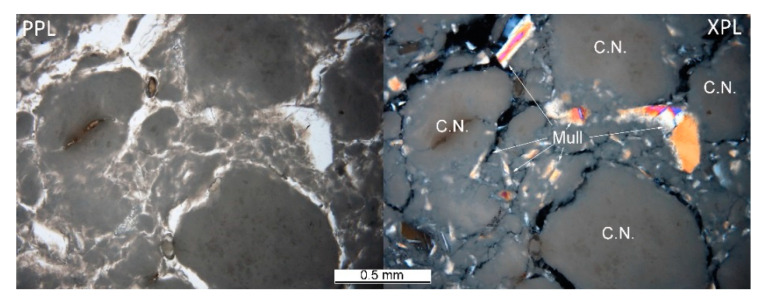
Sample K-T–fired clay nodules surrounded by mixture of fired clay and mullite; PPL = plane polarized light, XPL = crossed polarized light.

**Figure 4 materials-14-00779-f004:**
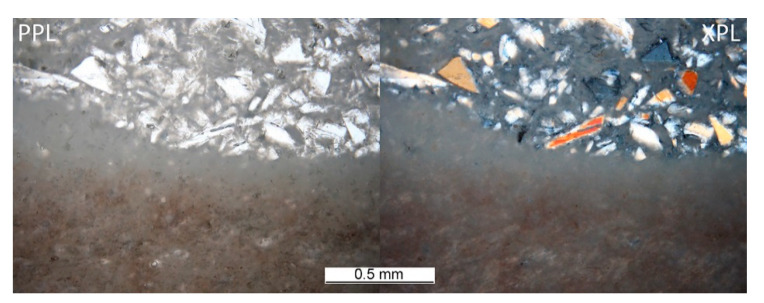
Sample G-T–interface of large, fired clay nodule and mixture of fired clay and fragments of fused mullite.

**Figure 5 materials-14-00779-f005:**
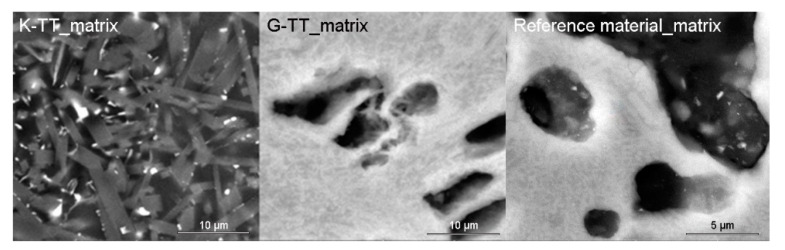
Comparison of mullite crystals size in matrices of K and G samples after two cycles of industrial firing with the reference material (firing temperature 1480 °C/5h soaking).

**Figure 6 materials-14-00779-f006:**
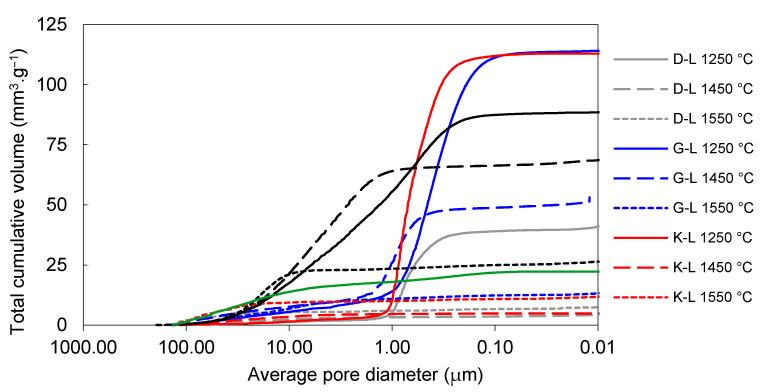
Total cumulative pore volume of laboratory prepared samples and reference material.

**Figure 7 materials-14-00779-f007:**
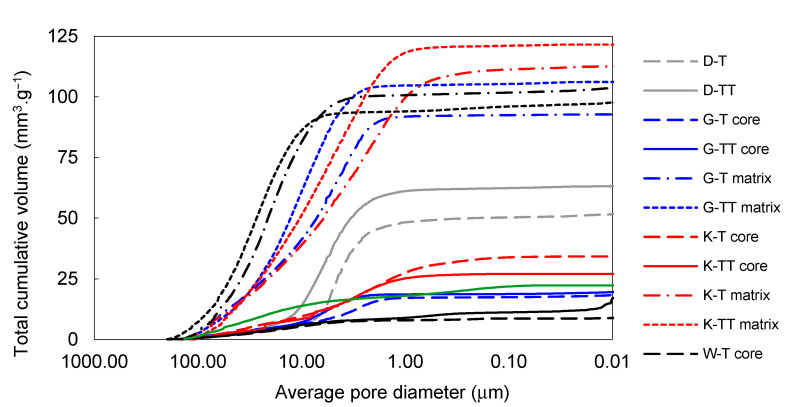
Total cumulative pore volume of samples prepared by pilot industrial tests and the reference material.

**Table 1 materials-14-00779-t001:** List of raw materials.

Raw Material	Claystone	Kaolin			Mullite Grog
	W Super	KN-83	GP3	DS1	MOTIM
Site/	Kaznějov site, Czech Republic/	Kosyatin, Ukraine/	Kosyatin, Ukraine/	Chlumčany site, Czech Republic/	Electrocorundum Ltd.
Producer	Keramost company	SOKA company	SOKA company	Sedlecký kaolin, a.s. company	Mosonmagyaróvár, Hungary

**Table 2 materials-14-00779-t002:** Calculated chemical compositions of raw mixes (85% clay, 15% MOTIM mullite) based on WDXRF analysis in wt.%, rounded to two decimal places.

Formula Designation	Raw Materials	Al_2_O_3_	SiO_2_	Fe_2_O_3_	TiO_2_	CaO	MgO	K_2_O	Na_2_O
W	W super claystone + MOTIM	48.62	47.51	0.88	1.37	0.37	0.15	0.49	0.25
K	KN83 kaolin + MOTIM	49.74	47.82	0.40	1.23	0.30	0.07	0.31	0.14
G	GP3 kaolin + MOTIM	48.53	48.86	0.76	0.67	0.26	0.20	2.89	0.10
D	DS1 kaolin + MOTIM	45.20	50.19	0.54	0.56	0.32	0.20	0.71	0.38

**Table 3 materials-14-00779-t003:** Particle size distribution of MOTIM mullite, 0–1 fraction.

**Sieve Opening [mm]**	>1	0.5–1.0	0.2–0.5	0.2–0.09	<0.09
**wt.%**	0	44	34.6	15	6.4

**Table 4 materials-14-00779-t004:** Chemical and phase composition of raw materials in wt.%, rounded to two and one decimal places respectively.

Raw Material/Composition	Claystone	Kaolin			Mullite Grog
	W Super	KN-83	GP3	DS1	MOTIM
Loss of Ignition (1000 °C)	−14.24	−13.47	−13.15	−12.46	
Chemical Composition					
Al_2_O_3_	42.15	43.53	42.04	37.93	76.38
SiO_2_	53.19	53.57	54.86	56.49	23.40
Al_2_O_3_:SiO_2_ ratio	0.79	0.81	0.77	0.67	3.26
Fe_2_O_3_	1.03	0.43	0.87	0.61	0.01
TiO_2_	1.68	1.50	0.81	0.68	0.01
CaO	0.38	0.29	0.24	0.12	0.01
MgO	0.17	0.07	0.23	0.24	–
K_2_O	0.85	0.36	0.57	3.53	0.08
Na_2_O	0.45	0.15	0.28	0.1	0.1
Phase Composition and General Formula					
Kaolinite (Al_2_Si_2_O_5_(OH)_4_)	80.2	93.3	89.7	69.9	–
Illite (K_0.65_Al2(AlSi_3_O_10_)(OH)_2_)	11.7	4.1	4.7	6.2	–
Smectite (no general formula available)	2.8			–	–
Anatase (TiO_2_)	2.5	0.8	0.7	0.6	–
Quartz (SiO_2_)	2.8	1.8	4.3	10.8	0.1
Feldspars (K_x_Na_y_Ca_1−(x+y)_Al_2−(x+y)_Si_2+(x+y)_O_8_)	-	-	0.6	12.5	–
Mullite (Al(4 + 2x)Si(2 − 2x)O(10 − x) where x = 0.17 to 0.59)	–	–	–	–	80.5
Corundum (Al_2_O_3_)	–	–	–	–	0.2
Cristobalite (SiO_2_)	–	–	–	–	
Amorphous phase	NA	NA	NA	NA	19.2

**Table 5 materials-14-00779-t005:** Phase composition of prepared grog samples (L = laboratory oven, T = industrial tunnel kiln one firing cycle, TT = industrial tunnel kiln two firing cycles) and reference material–Reference material. The formulas with similar composition as the reference material are highlighted by red colour.

Sample	Firing	Mullite	Corundum	Cristobalite	Quartz	Amorphous Phase
Designation	Temperature [°C]					
WL	1250	74.5	0.0	16.1	4.0	5.5
	1450	75.6	0.0	7.4	3.8	13.1
	1550	71.2	0.0	0.5	0	29.7
W-T	1550	58.5	0.5	20.3	0.9	19.8
W-TT	1550	54.9	0.7	14.6	0.4	29.4
G-L	1250	69.5	0.0	19.6	3.0	7.9
	1450	70.7	0.0	24.8	1.0	3.5
	1550	71.7	0.0	0.6	0.0	27.7
G-T	1550	56.9	0.1	21.8	0.1	21.1
G-TT	1550	55.9	0.1	23.2	0.0	20.8
K-L	1250	59.1	0.0	0.0	3.9	36.8
	1450	57.2	0.0	0.0	0.0	42.8
	1550	54.9	0.0	0.0	0.0	45.1
K-T	1550	67.2	0.0	31.1	0.1	1.6
K-TT	1550	66.5	0.0	24.4	0.0	9.1
D-L	1250	56.6	0.0	0.0	4.0	39.4
	1450	55.3	0.0	0.0	0.7	44.2
	1550	54.9	0.0	0.0	0.6	44.5
D-T	1550	51.2	0.0	1.6	0.0	47.2
D-TT	1550	49.5	0.0	1.8	0.1	48.6
Reference Material		70.1	2.4	21.1	0.3	6.0

**Table 6 materials-14-00779-t006:** Mean size of mullite crystals [µm] on matrix of laboratory-produced samples, MOTIM mullite and reference material based on SEM observation.

**Firing Temperature [°C]/Sample**	1250	1450	1550
W-L	0.2	0.6	4.0
G-L	0.2	1.0	6.0
K-L	0.5	2.0	10.0
D-L	0.1	0.4	2.0
**Commercial Mullite Based Products**			
Reference Material	3.0		
MOTIM	283.0		

**Table 7 materials-14-00779-t007:** Assumed total price of aggregates produced by pilot experiments compared to products available on the market.

Aggregate Designation	Price in €/t
G	320
K	330
D	310
W	240
Commercial Products	340–420

## Data Availability

Data available in a publicly accessible repository.

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
