# Peer review of "Development and Properties of New Mullite Based Refractory Grog"

_materials, 2021, doi:10.3390/ma14040779_

Round 1

Reviewer 1 Report

  1. Why was the CaF2 chosen as the internal standard to quantify the content of the amorphous phase at line 172-175?
  2. The mullite peaks’ values of FWHM were significantly higher in the samples fired at 1250 °C, this was due to the grain size of mullite was too small finally resulted the increasing of the width of diffraction peak. (line 240)
  3. The interfaces of these regions were relatively sharp (Fig. 4) with no substantial porosity possibly decreasing the physical-mechanical properties of the samples. Please explain this statement at line 278-290.
  4. The author indicated the Grey coloring of the nodules near the interfaces was caused by migration of iron oxides towards their central parts. Whether could TiO2 cause the same phenomenon considering the similar content between iron oxides and TiO2?
  5. The comparison about mullite grain size with different sample at Fig.5 was carried with the same sintering temperature. The sintering temperature should be marked at line 300.

Author Response

Dear reviewer
Thank you very much for your time and fruitful comments. We accepted some of our suggestions and recommendations and changed our manuscript. Here are our answers and explanations.

On behalf of all authors
Karel Dvořák

  1. Why was the CaF2 chosen as the internal standard to quantify the content of the amorphous phase at line 172-175? Despite obvious reason (cubic symmetry, not too many overlaps with the peaks of the original sample phases etc.), CaF2 was used as the internal standard because of its relatively low microabsorption of X-rays in comparison to commonly used zincite.
  2. The mullite peaks’ values of FWHM were significantly higher in the samples fired at 1250 °C, this was due to the grain size of mullite was too small finally resulted the increasing of the width of diffraction peak. (line 240). Yes, it was due to smaller crystallite size. This fact is commented in the previous sentence: “Based on the mullite peak profiles, it is apparent that the crystallinity of mullite increased between 1250 and 1450 °C.” We added the expression “crystallite size” to this sentence to make it more clear.
  3. The interfaces of these regions were relatively sharp (Fig. 4) with no substantial porosity possibly decreasing the physical-mechanical properties of the samples. Please explain this statement at line 278-290. Accepted. Physical-mechanical properties of the composite products are usually strongly affected by the interfaces of different used materials. (We added this sentence to the text.) Extended porosity would certainly decrease them.
  4. The author indicated the Grey coloring of the nodules near the interfaces was caused by migration of iron oxides towards their central parts. Whether could TiO2 cause the same phenomenon considering the similar content between iron oxides and TiO2? Thank you for the comment. We assume that TiO2 does not have such a significant colouring effect as iron oxides.
  5. The comparison about mullite grain size with different sample at Fig.5 was carried with the same sintering temperature. The sintering temperature should be marked at line 300. Accepted.

Reviewer 2 Report

Manuscript ID: materials-1065961

Title: Development and properties of new mullite based refractory grog

Authors: David Zemánek et al.

Introduction. Authors should write specific facts, not general words:

Line 46-47. What exactly has changed?

Line 54-55. What does unsatisfactory mean?

Line 46-56. This information has not any references.

Line 57-62. This information is not correlate to the Title of article.

Line 81-82. Add value for these characteristics.

Line 84. How much cheaper?

Line 85. What’s temperature?

Line 96. Add value for these characteristics.

Line 102. Add full formulas for andalusite, sillimanite and kyanite.

Table 3. What is the particle size distribution of kaolin KN83, GP3, DS1?

Table 4. Add full formulas for all minerals.

The data in Figure 2 and Table 5 are identical. Delete one so that there is no repetition of information.

Figure 3-4. Why do the authors show only 2 samples out of 4? At what temperatures were these samples obtained?

Table 6. How was calculated this data? Add this information to section 2.

Table 7. Authors must describe the calculation in detail with equations. Authors must add reference for commercial products price.

The references contain only 11 sources, some of which are very old. Authors must add to the list of references more than 20 sources and use articles from 2019-2020.

For the text clarity would you refrain from using additional words, mostly meaningless words, which can be omitted, or some archaic words see e.g. "Therefore", "However", "Respectively", "Moreover", "Furthermore", "Nonetheless", "Perceivably," etc.

Reviewer's conclusion:

The article is interesting and is devoted to reducing the cost of production of refractory materials based on mullite. Many experiments have been carried out. An attempt has been made to calculate the cost of new types of refractory products. However, the introduction section requires significant improvement. The results of XRD and SEM analyzes of samples should be added to the main part of the article along with a discussion of the obtained results. It is necessary to substantially supplement the economic part of the article and add new sources to the list of references.

In this form, the article requires major revision before final acceptance.

Author Response

Dear reviewer
Thank you very much for your time and fruitful comments. We accepted some of our suggestions and recommendations and changed our manuscript. Here are our answers and explanations.

On behalf of all authors
Karel Dvořák

Reviewer 2 (Line numbering refers to the original text befor corrections)

Introduction. Authors should write specific facts, not general words. Accepted. We omitted the genaral words, where they are non necessary.

Line 46-47. What exactly has changed? The situation is explained in the Introduction part („The general situation in the refractory industry has changed significantly during the last decades (especially in recent years), which also affected the production of fired clay. Along with drop in demand for standard fired clay caused by changes in the steel manufacturing technology, the production of fired claystone has decreased.”)

Line 54-55. What does unsatisfactory mean? Accepted. We added a short explanation to the paper. (After the first attemptes the samples did not have the wanted shape and size. Also the product was not copmpletely and properly fired.)

Line 46-56. IThis information has not any references. Accepted. Appropriate references were added to this paragraph.

Line 57-62. This information is not correlate to the Title of article. Accepted. The this part was omitted.

Line 81-82. Add value for these characteristics. Accepted. This paragraph was shortened.

Line 84. How much cheaper? Accepted. This paragraph was shortened. Some of the prices are given in the Results and discussion chapter.

Line 85. What’s temperature? Accepted. This paragraph was shortened.

Line 96. Add value for these characteristics. Accepted.

Line 102. Add full formulas for andalusite, sillimanite and kyanite. The full formula of all these three minerals is identical: Al2SiO5. We omitted the the expression „type“ from the sentence to make it more clear.

Table 3. What is the particle size distribution of kaolin KN83, GP3, DS1? All the kaolins were floated. We added this information to the text. The floating procedure limits the grain size distribution.

Table 4. Add full formulas for all minerals. Accepted. We added mineral formulas wehere it was possible. This is rather complicated, because there is no uniform formula for groups of minerals such as smectites and feldspars. Also formulas of more simle minerals may vary.

The data in Figure 2 and Table 5 are identical. Delete one so that there is no repetition of information. Thank you for the comment. You are rigt, the data on mullite content are doubled. Figure 2 synopticly shows the contents of mullite only, which we consider a crutial information. Table 5 gives the complete quantitative phase analysis results (including mullite content). If it is not a serious problem, we prefer to keep it as it is.

Figure 3-4. Why do the authors show only 2 samples out of 4? At what temperatures were these samples obtained? The opticly observable microstructure was the same in all the examined samples. We added this informatin to the text.

Table 6. How was calculated this data? Add this information to section 2. Accepted. The size of mullite crystals was observed by SEM and measured along with crystallographic c axes. We added this information.

Table 7. Authors must describe the calculation in detail with equations. Authors must add reference for commercial products price. Accepted. The equation was added. The commercial products were find out by personal communication.

The references contain only 11 sources, some of which are very old. Authors must add to the list of references more than 20 sources and use articles from 2019-2020. Accepted. Ten relevant references were added.

For the text clarity would you refrain from using additional words, mostly meaningless words, which can be omitted, or some archaic words see e.g. "Therefore", "However", "Respectively", "Moreover", "Furthermore", "Nonetheless", "Perceivably," etc. Accepted.

Reviewer's conclusion:

The article is interesting and is devoted to reducing the cost of production of refractory materials based on mullite. Many experiments have been carried out. An attempt has been made to calculate the cost of new types of refractory products. However, the introduction section requires significant improvement. The results of XRD and SEM analyzes of samples should be added to the main part of the article along with a discussion of the obtained results. Thank you for this suggestion. In our oppinion, this would make the paper less comprehensive, since the graphical XRD results and SEM images take two and half pages. It is necessary to substantially supplement the economic part of the article and add new sources to the list of references.

Reviewer 3 Report

Dear Editor,

I have reviewed this paper with close inspection. The topic of the present publication is very interesting, and I have the following comments for the authors before its publication: 

  1. The abstract is very confusing for the reader, at least at its beginning, and it must be corrected.
  2. The introduction is very extended with a lot of extra details that are not necessary. Please reduce it in order to have a better flow between the paragraphs.
  3. Please rewrite lines 224-226.
  4. In the text, the authors use the names of samples as W, D, K, etc. but they do not analyze which sample is each of them clearly. Please add it somewhere!
  5. 1, please modify the y’ axis title in order to be obvious on the graph where the wt. % is referred.
  6. It is better to refer o semi-quantitative analysis instead of quantitative-based on XRD analysis.
  7. If the authors can calculate the samples' mullite crystal size with one more technique, the quality of the paper will increase.
  8. In Fig. 6 and Fig. 7 please remove the grid where it is possible as there are a lot of patterns and become confusing.
  9. Despite the big introduction, the references that the authors used are very limited. Please add some more important references.
  10. A revision in English is needed.

Author Response

Dear reviewer
Thank you very much for your time and fruitful comments. We accepted some of our suggestions and recommendations and changed our manuscript. Here are our answers and explanations.

On behalf of all authors
Karel Dvořák

  1. The abstract is very confusing for the reader, at least at its beginning, and it must be corrected.
  2. The introduction is very extended with a lot of extra details that are not necessary. Please reduce it in order to have a better flow between the paragraphs.
  3. Please rewrite lines 224-226.
  4. In the text, the authors use the names of samples as W, D, K, etc. but they do not analyze which sample is each of them clearly. Please add it somewhere! The formulas of the raw mixes (W, K, G, D) are given in table 2. The parametres of firing are given in table 5.
  5. 1, please modify the y’ axis title in order to be obvious on the graph where the wt. % is referred.
  6. It is better to refer o semi-quantitative analysis instead of quantitative-based on XRD analysis. We completely agree that the accuracy of the phase analysis results conducted by XRD is limited. In our oppinion, the expression „semiquantitative“ refers to the analysis, where not all present phases were quantified. We quantified all the identified phases includin the X-ray amorphous one. Therefore we use the expression „quantitative“ in the paper.
  7. If the authors can calculate the samples' mullite crystal size with one more technique, the quality of the paper will increase. Thank you for the comment. The only other accessible technique was light microscopy. Unfortunately, it was useless due to too small sizes of mullite – in may cases below the resolution of optical microscope.
  8. In Fig. 6 and Fig. 7 please remove the grid where it is possible as there are a lot of patterns and become confusing.
  9. Despite the big introduction, the references that the authors used are very limited. Please add some more important references. Ten relevant references were added.
  10. A revision in English is needed. English was checked.

Round 2

Reviewer 2 Report

The authors responded in detail to all reviewers comments.
In some cases authors simply deleted text and did not correct the introduction section. The task of the reviewer is to give advice on improving the quality of the article. Simple removal of information is not the right way out of the situation. However, this is authors choice.
Other changes have improved the quality of the article.

One aspect remains highly controversial: economic calculation.
The authors presented one equation by which the cost of the product was calculated. Using only one equation is not the right approach. I understand that the authors received some information from production representatives by personal communication. However, if authors declare an economic calculation, then it is necessary to describe the equations in more detail, add tables, etc. The article will look good even without economic comparison. Now, only additional questions arise to authors. Good economic calculation can itself be a subject of research. I ask the authors to think about future research with a deeper economic calculation.

All this does not decrease the scientific value of the study. The article can be accepted in this form.

Author Response

Thank you for the your additional comments. We are glad your final decision is positive.

We shortened the introduction part, since the reviewers had agreed we should do so and we thought it was correct. We also added ten new references to improve it. We belive that the introduction section gives the information relevant to the presented research.

We fully agree that the economic calculations represent a complicated task. One of the problems are business data which the producers do not want to publish for clear reasons. And, of course, these are rather estimates than exact calculations. However, we are convinced that the economic part of the paper will be very interesting for the readers.